# Fully automated, sequential focused ion beam milling for cryo-electron tomography

**Tobias Zachs[1], Andreas Schertel[2], João Medeiros[1], Gregor L Weiss[1], Jannik Hugener[1,3], Joao Matos[3], Martin Pilhofer[1]***

[1]Institute of Molecular Biology and Biophysics, Eidgenössische Technische Hochschule Zürich, Zürich, Switzerland; [2]Carl Zeiss Microscopy GmbH, Zeiss Customer Center Europe, Oberkochen, Germany; [3]Institute of Biochemistry, Eidgenössische Technische Hochschule Zürich, Zürich, Switzerland

**Abstract** Cryo-electron tomography (cryoET) has become a powerful technique at the interface of structural biology and cell biology, due to its unique ability for imaging cells in their native state and determining structures of macromolecular complexes in their cellular context. A limitation of cryoET is its restriction to relatively thin samples. Sample thinning by cryo-focused ion beam (cryoFIB) milling has significantly expanded the range of samples that can be analyzed by cryoET. Unfortunately, cryoFIB milling is low-throughput, time-consuming and manual. Here, we report a method for fully automated sequential cryoFIB preparation of high-quality lamellae, including rough milling and polishing. We reproducibly applied this method to eukaryotic and bacterial model organisms, and show that the resulting lamellae are suitable for cryoET imaging and subtomogram averaging. Since our method reduces the time required for lamella preparation and minimizes the need for user input, we envision the technique will render previously inaccessible projects feasible.

**\*For correspondence:**
pilhofer@biol.ethz.ch

## Introduction

Cryo-electron tomography (cryoET) is a powerful imaging technique at the interface of cell biology and structural biology, due to its capabilities for imaging cells in a near-native state and determining structures of macromolecular machines in their cellular context (*Beck and Baumeister, 2016*; *Koning et al., 2018*; *Kooger et al., 2018*; *Oikonomou and Jensen, 2017*; *Plitzko et al., 2017*). Unfortunately, cryoET is restricted to samples that are well below 800 nm in thickness, and requires sample thinning techniques for specimens like mammalian cells, *C. elegans*, yeast, cyanobacteria and biofilms. Biological cryoFIB milling is an emerging sample thinning technique, which uses a gallium ion beam to ablate segments of the sample in order to generate thin lamellae that can be imaged by cryoET (*Marko et al., 2007*; *Rigort et al., 2010*). Unlike previous methodologies, cryoFIB milling produces specimens without observable artifacts, in which *in situ* structural information is preserved. To date, its use has provided important insights into the cellular mechanisms of organisms too thick for direct imaging (e.g. *Ader et al., 2019*; *Albert et al., 2017*; *Böck et al., 2017*; *Bykov et al., 2017*; *Cai et al., 2018*; *Chaikeeratisak et al., 2019*; *Delarue et al., 2018*; *Khanna et al., 2019*; *Mahamid et al., 2019*; *Mahamid et al., 2016*; *Rast et al., 2019*; *Swulius et al., 2018*; *Weiss et al., 2019*). Unfortunately, however, cryoFIB milling for cryoET is at an early stage of technical maturation and the available techniques are highly manual procedures with relatively low throughput.

In current lamella preparation workflows (*Marko et al., 2007*; *Medeiros et al., 2018*; *Rigort et al., 2010*; *Strunk et al., 2012*; *Zhang et al., 2016*), samples are vitrified on transmission electron microscopy (TEM) grids by plunge-freezing. Grids are then transferred to a FIB-scanning

electron microscope (SEM) instrument, where potential targets are identified by SEM and FIB imaging (*Figure 1—figure supplement 1A/B*). Using a series of 'rough milling' steps, sections above and below the desired lamella are sequentially removed by reducing the separation between two milling areas and using decreasing FIB milling currents (e.g. 700 to 100 pA) (*Figure 1—figure supplement 1C–E*). Once the lamella is thinned to ~500 nm, additional targets are identified and thinned by rough milling in the same manner. To generate lamellae with a final thickness of 100–250 nm, the user returns to each target location and thins ('polishes') each lamella using a low (e.g. ≤50 pA) current (*Figure 1—figure supplement 1F*).

Using this method, up to 16 lamellae can be generated in a 10 hr session (*Medeiros et al., 2018*). However, for such a session, the procedure requires constant attention from the operator. This includes, visually monitoring milling and providing manual inputs every 5–15 min, for example to execute a series of repetitive tasks such as target identification, positioning milling patterns, changing FIB currents, monitoring drift and visually determining milling end points. This results in a strenuous procedure with a low throughput relative to the time invested by the user, as well as idle times due to input delays from the operator. To overcome these issues, automated sequential cryo-FIB milling has become of paramount interest for the field.

## Results

### Setup of a sequential automated milling session

Complementary to efforts from the de Marco and Raunser/Plitzko labs (*Buckley et al., 2020*; *Tacke et al., 2020*), here we report an automated sequential FIB milling method aimed at preparing lamellae for subsequent cryoET imaging. Automation was implemented on two separate Zeiss Crossbeam 550 FIB-SEM instruments, using routines that are available in the SmartFIB software package (Zeiss Microscopy GmbH, Oberkochen, Germany). Particularly important are the modules for stage backlash and drift correction, which are critical for reliable targeting of lamella preparation sites. This allows the user to set up all milling targets and then execute milling in an unattended, fully automated manner.

To begin an automated milling session, FIB current alignments are verified to ensure accurate milling (*Figure 1A*). Grids are then loaded into the FIB-SEM instrument. To simplify navigation and target identification, an overview image of the SEM grid is captured and linked to the stage coordinates as described in the methods. Using the overview image for stage navigation, the first milling site is identified and centered in both the SEM and FIB views (*Figure 1B*). To ensure accurate targeting of the milling site, mechanical stage movement errors were reduced by implementing backlash correction for all autonomous stage movements. Next, a series of operations is executed before saving the targets final position (*Figure 1C–E*). First, stage backlash correction is manually executed and the target is re-centered in the FIB image (*Figure 1C*). Second, the target's stage coordinates are saved to the stage navigation menu. Third, the stage is manually moved off-target and autonomously returned to the saved target location (*Figure 1D*). In case the target is *not* properly centered, the above three steps are repeated (*Figure 1E*), otherwise the user can proceed.

Next, patterns with specific currents for rough milling (e.g. 700, 300 and 100 pA) and polishing (e.g. 50 pA) are manually placed onto the target's FIB image (*Figure 1F/F'*). This is achieved by either generating a new set of patterns with user-defined pattern size, milling depth, milling current and material type; or by choosing a previously designed set of patterns, to reproduce a milling approach and decrease the needed setup time. To further improve the accuracy of targeting, we also incorporated an additional targeting step based on drift correction (*Figure 1F/F'*) for both rough milling and polishing. To implement this, each set of milling patterns receives a drift correction box, with user-defined dimensions, which is manually placed in a location close to the target. By capturing and saving an image of the area encompassed by the drift correction box, the milling patterns are anchored to their positions on the target.

After separately saving the first target's rough milling and polishing patterns to the queue, further targets are added by repeating the described procedure. This setup process takes ~9 min per target.

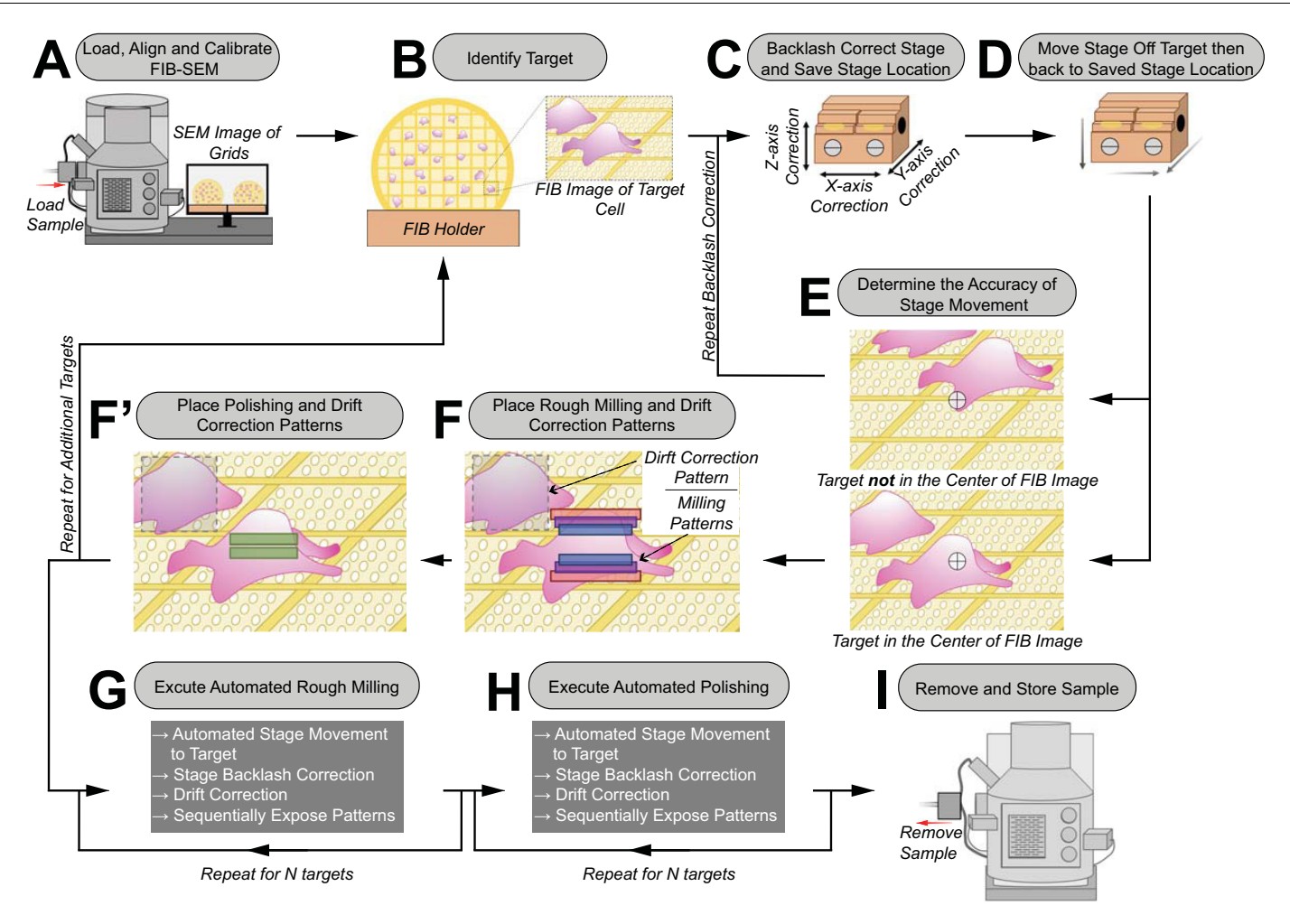

**Figure 1.** Schematic of the automated sequential cryoFIB milling workflow. (**A**) FIB currents are aligned and calibrated, and the sample is loaded into the FIB-SEM instrument. (**B**) A target cell is identified on the grid with the SEM and FIB. (**C**) To correct for errors in mechanical stage movements, backlash correction of the stage is performed. The resulting stage location is saved in the stage navigator. (**D**) The stage is randomly moved out of position by the user. Using the saved coordinates in the stage navigator, the stage is autonomously moved back to the target. (**E**) The accuracy of this autonomous stage movement is determined by the user. If the target is not centered in the FIB image, backlash correction is repeated until accurate targeting is achieved (**C–E**). (**F/F'**) Rough milling, polishing and drift correction patterns are placed onto the image. Rough milling and polishing patterns are saved separately to the queue. The procedure (**B–F'**) is repeated to select additional targets. (**G/H**) Rough milling and lamellae polishing are executed automatically. (**I**) The grids with milled lamellae are removed and stored.

The online version of this article includes the following figure supplement(s) for figure 1:

**Figure supplement 1.** Schematic of the manual cryoFIB milling workflow.

## Processes during a sequential automated milling session

To begin sequential automation, exposure of rough milling patterns saved in the queue is initiated (*Figure 1G*). For each set of patterns, the stage automatically moves to the target position and executes stage backlash correction. Next, image shifts are determined between the drift correction image that was recorded during the setup procedure and a drift correction image that is recorded after arriving at the target location. Any existing shifts are compensated for, using FIB beam shifts, to achieve precise milling at the target location. The rough milling patterns are then exposed, from the highest to the lowest current. Previously, manual milling methods used a real-time view in order to visually determine the time that a FIB current needed to cut through the specimen by visual end point detection. In our automated approach, the exposure time is calculated by the software using a user-specified milling depth, milling current, pattern size and material type. After exposing the

**Table 1.** Overview and success rates of milling sessions.

| Session | Sample Type | Rough Milling | Rough Milling Success | Polishing | Polishing Success | Instrument # |
|---|---|---|---|---|---|---|
| A | Cyanobacteria | Automated | 10/10 | Manual | 10/10 | 1 |
| B.1 | Yeast | Automated | 5/5 | Manual | 5/5* | 1 |
| B.2 | Cyanobacteria | Automated | 5/5 | Automated | 5/5* | |
| C | Yeast | Automated | 20/20 | Automated | 11/20† | 1 |
| D | Cyanobacteria | Automated | 7/7 | Automated | 7/7 | 1 |
| E | Cyanobacteria | Automated | 7/7 | Automated | 7/7 | 1 |
| F | Cyanobacteria | Automated | 18/19‡ | Automated | 16/18 | 1 |
| G | Yeast | Automated | 10/10 | Automated | 10/10 | 2 |
| H | Yeast | Automated | 20/20 | Automated | 18/20 | 2 |
| I | Yeast | Automated | 19/20§ | Automated | 19/19 | 2 |
| | | Total Automated | 121/123 (98%) | Total Automated | 93/106 (88%) | |

\* Rough milling of both B.1 and B.2 was performed in the same session.

† Lamellae were left in the instrument for 10h beore polishing, leading to lamellae bending and a lower success rate.

‡ User selected one target twice for rough milling, leading to a failure in rough milling.

§ User accidentally milled a target on the grid bar rendering it unusable.

rough milling patterns for the first target, the procedure is automatically repeated for the remaining targets in the queue.

Subsequently, the user can decide whether to perform polishing for all targets in a manual or automated manner (*Figure 1H*). The automation of polishing follows the same routine as described above.

## Application of sequential automated milling

During the development of this method, we tested automated sequential milling on two separate Zeiss Crossbeam 550 instruments using the model organisms *Saccharomyces cerevisiae* strain SK1 (hereafter 'yeast') and the multicellular cyanobacteria *Anabaena* sp. PCC 7120 (hereafter 'cyanobacteria') in nine independent milling sessions (sessions A-I, *Table 1*). In these sessions, we aimed to generate between 5 and 20 lamellae (*Figure 2*). Rough milling success, as defined by the presence of an intact lamella at the targeted location after rough milling, was 98% (n = 123). The only failures in lamellae production were the result of user error, when rough milling was accidentally executed on the same target twice and on a grid bar (session F and I, respectively). With sessions B.1 and B.2, we also demonstrated the robustness of the targeting routine by successfully generating 10 lamellae spread across two grids containing two different samples (*Table 1*).

While these results present a significant step forward in TEM lamellae preparation, we next set out to implement automated sequential lamella polishing. In a series of sessions (B.2-I), we milled between 5 and 20 targets. In total, the success rate (intact lamella detected after polishing) of automated sequential polishing was 88% (n = 106 rough lamellae). Importantly, 9 of the 13 failed polishing attempts occurred in session C in which the rough-milled lamellae were left in the FIB-SEM instrument for 10 hr before beginning automated polishing. Prior to polishing, these rough lamellae showed signs of bending, which likely resulted in failure in lamellae polishing. In fact, when we repeated session C without a delay between rough milling and polishing (sessions H and I), we did not observe bending and obtained 95% polishing success. Therefore, we find it advisable to execute rough milling and polishing in quick succession.

## Assessment of sample quality

In order to assess sample quality, we transferred the grids from all sessions to the cryoTEM. Of the lamellae that were generated in a fully automated manner (n = 93), 16% were lost in transfer and 3% exhibited contamination or cracks prohibiting data collection. From the collected tomograms, we

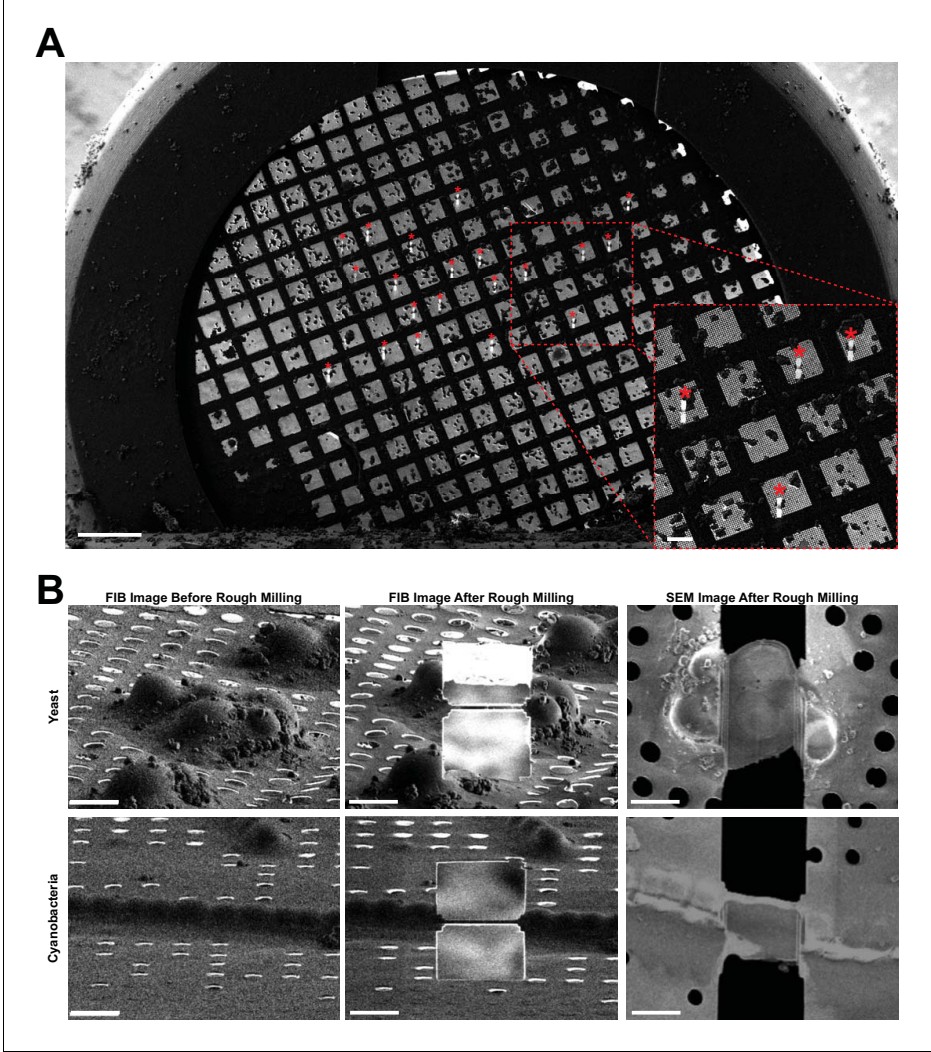

**Figure 2.** Representative images of lamellae generated by automated sequential rough milling. (**A**) SEM grid overview image including 20 yeast targets (asterisks) on which rough milling was performed in an automated sequential manner (session C). Bars, 200 μm. (**B**) Representative SEM and FIB images of yeast and cyanobacteria cells captured before and after fully automated sequential rough milling (session B.1 and B.2). Bars, 5 μm. The online version of this article includes the following source data and figure supplement(s) for figure 2:

**Figure supplement 1.** Further examples of lamellae generated in a fully automated manner.
**Figure supplement 2.** Distribution of lamellae thicknesses.
**Figure supplement 2—source data 1.** Source data for *Figure 2—figure supplement 2*.

determined the lamellae thicknesses to range from 117 to 379 nm when final polishing patterns are placed 300 nm apart (*Figure 2—figure supplement 1* and *Figure 2—figure supplement 2*). A possible contributor to the range of lamellae thicknesses is minute drift during polishing. The average thickness of lamellae generated in a fully automated manner (session B.2-I; 243 nm) is comparable to the thickness of manually polished lamellae (session A/B.1; 258 nm).

CryoET imaging of the lamellae generated during the automated session revealed distinct cellular features and macromolecular complexes (*Figure 3*). Yeast tomograms contained nuclei, nuclear pore complexes, mitochondria, endoplasmic reticula, cytoplasmic ribosomes and vacuoles. In one particular tomogram (*Figure 3B/C*), we observed microtubules inside the nucleus and were able to determine the number of protofilaments without averaging. Cyanobacterial tomograms showed thylakoid membranes, phycobilisomes and septal junctions (*Figure 3E*). To further assess sample and data quality, we performed subtomogram averaging of cyanobacterial septal junctions, which were

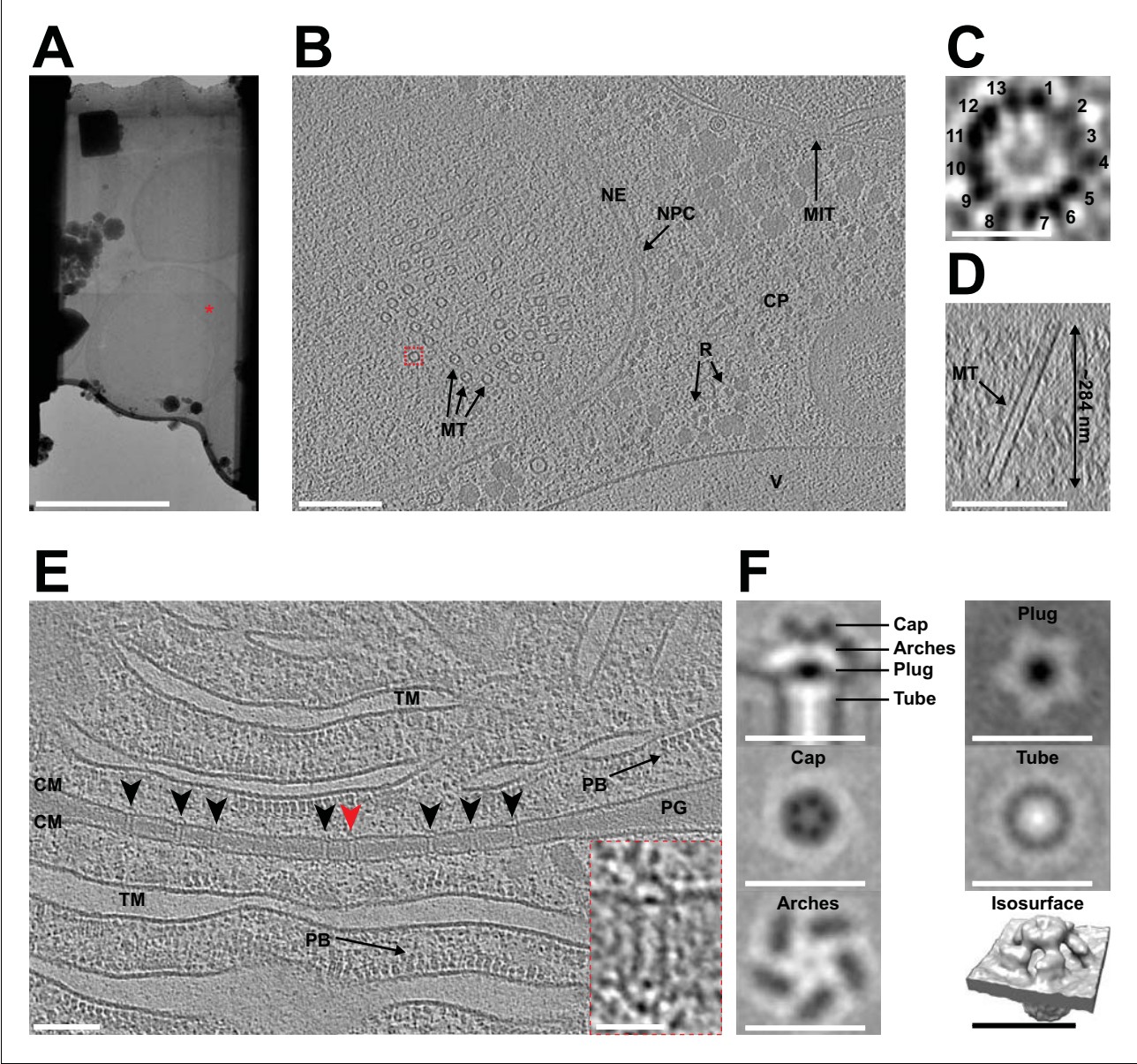

**Figure 3.** Automated sequential cryoFIB milling results in high-quality lamellae and cryotomograms. (A) CryoTEM overview image of a lamella (session H) containing three yeast cells. Red mark indicates the area imaged in (B). Bar, 5 µm. (B) A 22.85 nm thick slice through a cryo-tomogram of a yeast cell (session H) [indicated by red mark in (A)]. The tomogram shows a nuclear pore complex (NPC), nuclear envelope (NE), microtubules (MT), cytoplasm (CP), cytoplasmic ribosomes (R), mitochondria (MIT) and a putative vacuole (V). Bar, 200 nm. (C) Top view of the microtubule indicated by dashed red box in (B). From a single slice through the tomogram it is possible to identify 13 protofilaments that make up the microtubule. (D) Cross-section of the tomogram showing the microtubule in (C) inside the generated lamella. From this view, we also determined the lamella thickness to be ~284 nm. (E) Shown is a 14 nm thick slice through a cryo-tomogram of a septum between two cyanobacteria cells (session F). The thickness of the lamella was determined to be ~208 nm. Arrowheads indicate septal junctions. The inset shows a magnified view of the septal junction indicated by a red arrowhead. Other cellular features include cytoplasmic membranes (CM), phycobilisomes (PB), thylakoid membranes (TM) and septal peptidoglycan (PG). Bars, 100 nm and 25 nm (inset). (F) Subtomogram average generated by extracting 343 septal junction particles from nine tomograms and performing fivefold symmetrization. Shown are longitudinal and perpendicular slices (thickness 0.68 nm) and a surface rendering of the symmetrized average. The observed characteristic structural modules were similar to a recent study that applied manual cryoFIB milling (*Weiss et al., 2019*) (also see *Figure 3—figure supplement 1*). Bars, 25 nm.

The online version of this article includes the following source data and figure supplement(s) for figure 3:

**Figure supplement 1.** Comparison of data quality between manual and automated milling.
**Figure supplement 1—source data 1.** Source data for *Figure 3—figure supplement 1*.

characterized recently by a manual cryoFIB milling/cryoET approach (*Weiss et al., 2019*). From nine lamellae, a total of 412 subvolumes were extracted, averaged and classified in order to remove mis-aligned particles. The 343 remaining subvolumes were then averaged and symmetrized. The resulting structure had the same architecture as previously described, including a cap module with five arches, a plug module and a tube module (*Figure 3F*). Fourier shell correlation (FSC) analyses indicate that the average had a resolution that is similar to a structure that was calculated using the same number of particles extracted from tomograms generated in a previous study by manual milling (*Weiss et al., 2019*; *Figure 3—figure supplement 1*).

## Discussion

In conclusion, our automated sequential cryoFIB milling method allows for the production of high-quality lamellae for cryoET imaging and will impact cryoFIB/cryoET projects in several ways. First, the time investment by the operator is significantly reduced from ~10 hr in a manual milling session to ~2.4 hr for an automated sequential milling session, assuming 16 targets are milled. Second, by removing the need for frequent user inputs and idle times, the minimum required machine time is reduced from 30 to 45 min (*Medeiros et al., 2018*) (i.e. 16 lamellae in 10 hr) to ~25.75 min (~9 min setup plus ~16.75 min milling) per lamella. Note that the stated values are dependent on key parameters including the sample type, sample thickness, lamellae sizes and stage stability. Instrument 2, for example, required an additional stage settling time (3 min) before polishing each target. Third, based on the robustness and customizable nature of the method, the procedure can be adapted to a wide range of samples and milling techniques (*Toro-Nahuelpan et al., 2020*; *Wolff et al., 2019*). Fourth, the automated procedure will allow the user to systematically explore novel milling methods by reusing uniform milling patterns. Fifth, the method can generally be combined with correlated approaches that allow for target pre-screening, for instance cryo-light microscopy or cryo-FIB-SEM volume imaging (*Eibauer et al., 2012*; *Gorelick et al., 2019*; *Koning et al., 2014*; *Schertel et al., 2013*; *Schorb et al., 2017*; *Sviben et al., 2016*; *Vidavsky et al., 2016*). Altogether, the development of automated sequential cryoFIB milling renders cryoET applicable to previously unfeasible projects.

## Materials and methods

### Overview of the equipment and workflow

The method was established and tested on two different Crossbeam 550 FIB-SEM instruments (Carl Zeiss Microscopy) equipped with copper band-cooled mechanical cryo-stages and integrated VCT500 vacuum transfer systems (Leica Microsystems). The detectors used included an InLens secondary electron detector for determining grid topology (Carl Zeiss Microscopy) and a SE2 detector for identifying milling targets and visually assessing the ice thickness (Carl Zeiss Microscopy). In our workflow, EM grids were prepared with the yeast strain *Saccharomyces cerevisiae* SK1 and the cyanobacterial strain *Anabaena* sp. PCC 7120, clipped into FIB milling Autogrids (ThermoFisher Scientific, Waltham, MA). These grids were then mounted onto a pre-tilted Autogrid holder (*Medeiros et al., 2018*) (Leica Microsystems GmbH, Vienna, Austria) using a VCM loading station (Leica Microsystems). With the VCT500 shuttle, the Autogrid holder was transferred to an ACE600 cryo-sputter coater (Leica Microsystems). Under cryogenic conditions the plunge-frozen sample were sputter-coated with a ~4 nm thick layer of tungsten. After sputter coating, the samples were transferred into the Crossbeam 550 using the VCT500 shuttle. In the Crossbeam 550, the gas injection system (GIS) was used to deposit an organometallic platinum precursor layer onto each grid. Automated sequential FIB milling was subsequently set up and executed. Sample preparation, plunge-freezing, Autogrid mounting, holder loading and vacuum cryo-transfer steps were executed similarly to what was described in *Medeiros et al. (2018)*. Any deviations to the previously published protocol are described below.

### Cell culture and plunge freezing

FIB milling tests were performed using the cyanobacterial strain *Anabaena* sp. PCC 7120 and the yeast strain *Saccharomyces cerevisiae* SK1 (*Table 2*). Yeast and cyanobacterial strain identities were

**Table 2.** Strains used in this work.

| Strain | Identifier | Genotype | Source or reference |
|---|---|---|---|
| *Anabaena* sp. PCC7120 | Cyanobacteria | Wild Type | *Rippka et al. (1979)* |
| *Saccharomyces cerevisiae* SK1 | Yeast | diploid, homozygous for *ndt80Δ::HIS3 ho::LYS2 ura3 leu2::hisG trp1::hisG his3::hisG* | This study |

determined according to standard microbiological procedures. The cyanobacterial cell line originates from *Rippka et al. (1979)*. The cyanobacterial strain was grown and prepared for FIB milling as previously described in *Weiss et al. (2019)*. The yeast strain was generated in this study. Meiotic yeast cells were cultured as previously described in *Wild et al. (2019)* and prepared for plunge freezing as described by *Medeiros et al. (2018)*, with minor modifications. Briefly, yeast cells were harvested and 4 μL of cell suspension, $OD_{600}$ ~1.5, was applied on glow-discharged copper EM grids (R2/2, Quantifoil, Großlöbichau, Germany). Grids were back-blotted twice for 3–5 s each time, at 4°C with 95% humidity and plunged into liquid ethane/propane using a Vitrobot (ThermoFisher).

## Equipment calibration

To ensure that automated sequential FIB milling was successful, the Crossbeam 550 was properly aligned. While the SEM column alignments are stable and non-essential during automated milling, the FIB alignment between different currents at a given voltage (30 kV for biological cryo-samples) should be checked and optimized. Typically, this calibration is done weekly or when deemed necessary and takes roughly 60 min to complete. In case of deviation, on-the-fly adjustments are possible on a loaded cryo-sample, however, standard calibration procedures are best performed on a silicon wafer due to its structural homogeneity, which allows better evaluation of the FIB beam shape. Once inserted into the chamber, the stage was tilted by 54° to be perpendicular to the FIB beam and then moved to the working distance (i.e. coincidence point). Using the 'spot' function in an unexposed sample region, the beam was focused to its spot size allowing it to burn a hole into the silicon. When the current is properly calibrated, the beam will produce a spot that is round with sharp edges. This was best seen when using a mixed signal of the InLens and SE2 detector. If a beam spot had imperfections, like a tailing edge, beam parameters including focus, stigmatism and aperture alignments need to be improved and saved. After optimizing these parameters for each current, all currents were aligned against the reference current. This was best performed by positioning an easily recognizable structure, like a burnt hole, in the middle of the reference current image and then centering this feature in each of the other currents. Finally, to ensure that the currents were properly aligned, a location is imaged by each current. If properly aligned, switching between currents should not lead to focus changes or beam offsets.

## Sample coating

To enhance conductivity and decrease the effects of charging, the plunge-frozen sample was coated with a ~4 nm layer of tungsten. This was autonomously executed by an ACE600 sputter coating program in an argon atmosphere (8.0E-3 mbar) and with a current of 90 mA. After inserting the holder into the FIB-SEM, a protection layer of organometallic platinum precursor was deposited onto each grid to minimize the effects of curtaining. For cold deposition of platinum precursor, the holder was moved 3 mm below the coincidence point and tilted to 20°. By positioning the GIS needle above each grid and opening the GIS for 45 s, a layer of platinum precursor was deposited onto the sample. Since the GIS needle was mounted at a similar angle as the FIB column, deposition of platinum occurred preferentially on the side of the cells where the FIB beam hits the sample, ensuring the best protection. For deposition under cryo-conditions, it is essential that the heating element of the GIS needle and reservoir are turned off to keep the system at room temperature (28°C).

## Stage registration

To assist in the identification of targets, overview images of an entire EM grid are taken. On the Zeiss Crossbeam 550, these images can be coupled to the stage navigation. To calibrate stage registration a high-resolution (4096 × 3072 pixels, 35x magnification) overview image was taken with the SE2 detector, which provided the best information for identifying targets inside the vitrified ice and determining ice thickness. This overview image was then loaded onto the stage navigation bar and registered by correlating three distinctive points on the image to their specific positions on the stage as observed in the live SEM view. After completion, double clicking on a desired target in the navigation bar automatically moves the stage to the location of interest. In addition, backlash correction was also included for all automated stage movements, using the user preference settings of the software SmartSEM (Carl Zeiss Microscopy).

## Defining milling materials

To permit unsupervised automation of lamellae production, the Crossbeam 550 was calibrated to mill a cross-section with a specified depth through the sample. To ensure proper milling, the system needs to be calibrated for a distinct 'material' so that the correct milling parameters like dose are applied during milling. For cryo-TEM lamella preparation, the material 'vitrified ice' was created using a dose calibration of 20 mC/cm$^2$ being equivalent to a milling depth of 1 μm in cross-section mode. No attempts were made to make sample specific material types, however, this would also be possible.

## Parameters for imaging and milling

For SEM imaging, voltages from 1.9 to 5 kV and a constant current of 28 pA were used. To capture SEM images, we most commonly used the InLens detector to obtain surface information of the sample. During FIB imaging, on the other hand, a fixed voltage of 30 kV and a low current (20 pA) was used. FIB images were usually captured by using the SE2 detector, which is less sensitive to imaging-induced charging. During automated sequential milling a total of four currents were used for rough milling (700 pA, 300 pA and 100 pA) and polishing (50 pA). For milling, we defined the patterns to be executed using bi-directional and cross-section cycle mode with a 10 μm milling depth. It is important to note that the milling parameters (milling current, pattern size, milling depth, etc.) are adjustable on-the-fly to optimize each set of patterns for the target location and desired milling strategy.

## Automated sequential FIB milling protocol

To generate high-quality lamellae, it was essential to prepare the FIB-SEM and sample for automated sequential milling. Preparations included checking and calibrating the FIB currents, coating the plunge-frozen sample with a layer of tungsten and organometallic platinum, and performing stage registration. Once these steps were executed, automated sequential milling was initiated by identifying and setting up milling targets.

The grid overview image in the stage navigator was used to identify a milling target. The identified target was then manually centered in the live FIB view with the aid of the SEM. To improve the accuracy of automated stage movements, backlash correction was performed manually and implemented for all automated stage movements. The target's stage coordinates were then saved in the stage navigator. To ensure that the instrument was able to perform targeting during automation, the stage was manually moved away from the target and then instructed to move back to its saved location. The target was located using the live FIB view and if it was >1 μm from desired location, it was manually centered again. If manual centering was required, the new stage location was saved and the instrument's ability to perform targeting was tested again. To ensure successful milling during automation, it was essential to refine the stage location until the stage was able to perform targeting successfully.

Once an accurate stage movement was achieved, milling patterns were placed onto a targets FIB image captured using SmartFIB. In SmartFIB, each pattern contains specific milling conditions (i.e. current, milling depth, size, shape, etc.) and a designated FIB milling location. SmartFIB allows the placing of multiple patterns with different conditions onto a single FIB image in order to perform automated milling. Patterns were placed and their properties were changed by using the SmartFIB

GUI in the 'attributes' tab. When testing this methodology, we placed eight rectangular milling patterns: six rough milling and two polishing patterns (*Table 3*). The final polishing patterns were spaced 300 nm apart, since we generally expected the generated lamellae to be thinner than the specified value. To make uniform lamellae, it was also possible to save these eight patterns as a recipe, which can be dragged and dropped onto images of other milling targets. To then save these milling patterns, it was essential to separate the rough and polishing patterns. This was accomplished by deleting the polishing patterns from our recipe, saving only the rough milling patterns, undoing the deletion of the polishing patterns (using the SmartFIB 'undo' button), deleting all rough milling patterns and then saving only the polishing patterns.

To improve the targeting accuracy of this methodology, a drift correction step was also added to each set of rough milling and polishing patterns immediately before being saved. This was done in the SmartFIB 'attributes' tab, by capturing and saving an image of a defined region of the FIB view. During the automated protocol SmartFIB would use this image to perform image recognition before beginning milling and compensate for small shifts (generally between 0 and 2 µm) to ensure the milling patterns are placed correctly on the target. When testing this methodology, it was important to use the same drift correction image for both the rough milling and polishing patterns. This ensured that the same region is exposed during rough milling and polishing.

After saving a set of rough and polishing patterns, the described method can be repeated for further targets. For an automated protocol, about 9 min were needed to set up each target. Once satisfied with the number of targets, all rough milling recipes in the SmartFIB queue were selected and exposed. Exposure of a typical rough milling target took about 12 min. Upon completion, rough milling targets were observed using the SEM and FIB to determine their quality. To then initiate polishing, it is possible to either tick all polishing recipes and expose them, or individually move to each target using SmartFIB, take a FIB image, manually drag polishing patterns into place and expose the lamella. Polishing typically took about 4.75 min. Once all targets are polished, the lamellae are removed from the instrument and stored. Note that instrument 2 (*Table 1*) suffered from stage drift, which caused issues during polishing. Unfortunately, SmartFIB does not have an integrated wait time, however, using a set of low current (1–5 pA) patterns, we were able to apply a wait time of 3 min to allow the stage to settle before polishing. An overview of all the milling attempts that were performed can be found in *Table 1*.

## Cryo-electron tomography, tomogram reconstruction and subtomogram averaging

Data was collected on a Titan Krios 300kV electron microscope (ThermoFisher) equipped with a field emission gun, imaging filter (Gatan, Pleasanton, U.S.) (slit width 20 eV) and K2 or K3 direct electron detector (Gatan). To generate an overview of each grid, grid montages were collected at 135x magnification using SerialEM (*Mastronarde, 2005*). Cyanobacteria data was collected with UCSF Tomo

**Table 3.** Dimensions and currents used for each milling pattern during automated sequential lamellae preparation.

| Milling Pattern # | Pattern Attributes |
|---|---|
| Milling pattern 1 and 2 (Rough Milling 1) | 30 kV 700 pA<br>9 × 5 µm rectangle<br>Generates 2 µm thick lamella |
| Milling Pattern 3 and 4 (Rough Milling 2) | 30 kV 300 pA<br>8 × 2 µm rectangle<br>Generates 1 µm thick lamella |
| Milling Pattern 5 and 6 (Rough Milling 3) | 30 kV 100 pA<br>7.5 × 1 µm rectangle<br>Generates 500 nm thick lamella |
| Milling Pattern 7 and 8 (Lamella Polishing) | 30 kV 50 pA<br>7 × 0.5 µm rectangle<br>Generates 300 nm thick lamella |
| Drift Correction Pattern (Rough Milling and Polishing) | 3 × 3 µm rectangle |

(*Zheng et al., 2007*) at 2° increments between +60° and −60°. Data was collected at a defocus of −8 μm, total accumulated dose of ~140 e⁻ / Å² and pixel size of 3.38 Å. Yeast tomograms were collected using SerialEM between +60° and −60° at 2° increments with a defocus of −8 μm, total accumulated dose of ~120 e⁻ / Å² and pixel size of 4.57 Å. Tomogram reconstruction and subtomogram averaging was performed according to *Weiss et al. (2019)*. Briefly, tomograms were reconstructed using the IMOD package (*Kremer et al., 1996*) and septal junction subtomogram averaging was performed using PEET (*Nicastro et al., 2006*). A total of 412 particles were extracted and averaged in a box of 44 × 44 × 44 pixels with a pixel size of 6.8 Å. PEET classification was then used to remove misaligned particles (343 final particles). 5-fold symmetry was applied to obtain the final average. The FSC (Fourier Shell Correlation) was generated by using the PEET command calcFSC.

## Acknowledgements

We thank Saskia Mimietz-Oeckler and Andreas Hallady (Leica Microsystems GmbH) for technical support. We thank ScopeM for instrument access at ETH Zürich. MP was supported by the Swiss National Science Foundation (#31003A_179255), the European Research Council (#679209) and the Nomis Foundation.

## Additional information

### Competing interests
Andreas Schertel: Is an employee of Carl Zeiss Microscopy GmbH. The other authors declare that no competing interests exist.

### Funding

| Funder | Grant reference number | Author |
| --- | --- | --- |
| Swiss National Science Foundation | #31003A_179255 | Martin Pilhofer |
| European Research Council | #679209 | Martin Pilhofer |
| NOMIS Foundation | | Martin Pilhofer |

The funders had no role in study design, data collection and interpretation, or the decision to submit the work for publication.

### Author contributions
Tobias Zachs, Andreas Schertel, João Medeiros, Conceptualization, Investigation; Gregor L Weiss, Jannik Hugener, Investigation; Joao Matos, Supervision, Funding acquisition; Martin Pilhofer, Conceptualization, Supervision, Funding acquisition, Investigation, Project administration

### Author ORCIDs
Tobias Zachs  https://orcid.org/0000-0002-0836-0989
João Medeiros  https://orcid.org/0000-0001-9075-548X
Joao Matos  https://orcid.org/0000-0002-3754-3709
Martin Pilhofer  https://orcid.org/0000-0002-3649-3340

### Decision letter and Author response
Decision letter https://doi.org/10.7554/eLife.52286.sa1
Author response https://doi.org/10.7554/eLife.52286.sa2

## Additional files

### Supplementary files
• Transparent reporting form

## Data availability

The tomography data are deposited at EMDB/EMPIAR. EMPIAR-10376, EMD-10707, EMD-10708, EMD-10710.

The following datasets were generated:

| Author(s) | Year | Dataset title | Dataset URL | Database and Identifier |
|---|---|---|---|---|
| Zachs T, Pilhofer M | 2020 | Tomograms | http://www.ebi.ac.uk/pdbe/entry/emdb/EMD-10707 | Electron Microscopy Data Bank, EMD-10707 |
| Zachs T, Pilhofer M | 2020 | Tilt Series | https://www.ebi.ac.uk/pdbe/emdb/empiar/entry/10376 | Electron Microscopy Data Bank, EMPIAR-10376 |
| Zachs T, Pilhofer M | 2020 | Tomograms | http://www.ebi.ac.uk/pdbe/entry/emdb/EMD-10708 | Electron Microscopy Data Bank, EMD-10708 |
| Zachs T, Pilhofer M | 2020 | Tomograms | http://www.ebi.ac.uk/pdbe/entry/emdb/EMD-10710 | Electron Microscopy Data Bank, EMD-10710 |

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
