## [Decision Letter]

**Acceptance summary:**

In this manuscript, an automated procedure for thinning vitreous cellular samples by cryo-focused ion beam milling is presented. Thinning is required to make all cell regions accessible to cryo-electron tomography, and thereby enable in situ structural studies, which are important for understanding cellular architecture on the molecular level. It has been a major bottleneck to produce sufficient numbers and a consistent quality of thinned cellular samples. This paper is thus a significant advancement for the use of cryo-FIB-milling for thinning and will therefore be of great interest to a broad community. The authors collaborated with an industrial partner (Zeiss) to implement automation. They quantified the improvement in throughput, and they provide evidence for consistency in quality of preparing yeast cells as well as cyanobacteria. They use the latter data set for subtomogram averaging of a septal structure that they have characterised in a previous paper where they generated lamellae manually, and they now show that data from automatically thinned cells result in a 3D average structure of comparable quality.

**Decision letter after peer review:**

Thank you for submitting your article "Fully automated, sequential focused ion beam milling for cryo-electron tomography" for consideration by *eLife*. Your article has been reviewed by three peer reviewers, and the evaluation has been overseen by a Reviewing Editor and John Kuriyan as the Senior Editor. The reviewers have opted to remain anonymous.

The reviewers have discussed the reviews with one another and were all strongly supportive of your manuscript. It serves as a necessary and welcomed advancement in automating cryo-FIB/SEM, not only increasing the throughput of this technique but also decreasing errors and decreasing operator frustration. The Reviewing Editor has drafted this decision to help you prepare a revised submission.

Essential revisions:

There are a few things that would make the manuscript much more useful to the wider community, and the reviewers and I would like to ask the authors to address them or include them before publication:

1) The entire manuscript has very few pictures of lamellae. It only shows two lamellae imaged by the SEM (one yeast, one Anabaena) and one by TEM (yeast). This is way too little for a manuscript describing automated making of lamellae. A mosaic of many lamellae carried out during at least one session should be included (ideally both SEM and TEM images), so that the expert user can gauge the quality of the lamellae made.

The authors have to make sure that the raw data (namely tilt series) is deposited to EMPIAR. Tomograms and subtomogram averages need to be submitted to the EMDB.

2) Along these lines, the authors state that all lamellae were imaged by cryo-ET. Were there any lamellae broken? Did they have curtaining or contamination? Statistics on the things that often limit throughput of lamellae production should be discussed.

3) It's a pity that the experiment in Session C was not successful due to the time the samples were left inside the microscope. This is valuable information and I am glad the authors included it. However, since it was the only automated experiment that included polishing not on the bacteria that the lab is well familiar with (and since technically it should take a day in the dual beam and a day in the TEM), this experiment should be repeated.

4) If there are clear published protocols for tungsten deposition, the authors should cite them, otherwise a slightly longer explanation is warranted both for users with the same instrument and others trying to adapt the process to the tools available to them.

5) If needed, could each lamella be independently adjusted for depth? Some samples have a lot of variability and the parameters chosen can widely differ. This also goes for the height of the initial boxes. The authors always use 10 μm milling depth. Was this value an upper bound of what the depth expected their samples to have? Would a smaller value lead to lamellae not breaking through, getting curtains, etc? In other words, how should the users select this value?

6) In many of the papers cited by the authors that use cryo-FIB-ET, the lamellae are much wider. While the automation will be very useful and is a much needed step in the workflow, current users can get the idea that it will increase the throughput by an order of magnitude. This is partially because the lamellae generated in this report are significantly smaller than those reported in many of the papers that the authors cite.

The typical lamellae width in many of those papers are at least 10-12 μm, and a similar or longer depth. So, a typical mammalian cell (and many yeast) lamellae have an area of over 150 μm^2. In contrast, the lamellae shown here (Figure 2), have areas of about half. Similarly, the total volume that is milled away with the initial patterns is much larger in mammalian cells. Mammalian cells are at least a few thousands of cubic microns in volume, cf. the volume of a single yeast cell, 50-100 μm3. Thus, a lamella containing 3-4 yeast cells as shown in this manuscript is an order of magnitude less in volume than a mammalian cell. Thus, in order to be able to compare to other work, and to adjust expectations by the reader, the authors should specify that the numbers they cite (even in their previous manual work) of up to 16 lamellae in 10 hours applies for large single bacteria or for very small sets of yeast cells. While their reporting is accurate, it would be better (and equally impressive) to compare any gains in time with respect to their previous work.

---

## [Author Response]

Essential revisions:There are a few things that would make the manuscript much more useful to the wider community, and the reviewers and I would like to ask the authors to address them or include them before publication:1) The entire manuscript has very few pictures of lamellae. It only shows two lamellae imaged by the SEM (one yeast, one Anabaena) and one by TEM (yeast). This is way too little for a manuscript describing automated making of lamellae. A mosaic of many lamellae carried out during at least one session should be included (ideally both SEM and TEM images), so that the expert user can gauge the quality of the lamellae made.

We have now generated Figure 3—figure supplement 1 that shows images of multiple lamellae.

The authors have to make sure that the raw data (namely tilt series) is deposited to EMPIAR. Tomograms and subtomogram averages need to be submitted to the EMDB.

Tilt series, tomogram and averages have now been uploaded. Accession numbers are stated under ‘Data and code availability.’

2) Along these lines, the authors state that all lamellae were imaged by cryo-ET. Were there any lamellae broken? Did they have curtaining or contamination? Statistics on the things that often limit throughput of lamellae production should be discussed.

We have included this information in the text: “Of the lamellae that were generated in a fully automated manner (n=93), 16% were lost in transfer, 3% exhibited contamination or cracks that prohibited data collection.”

3) It's a pity that the experiment in Session C was not successful due to the time the samples were left inside the microscope. This is valuable information and I am glad the authors included it. However, since it was the only automated experiment that included polishing not on the bacteria that the lab is well familiar with (and since technically it should take a day in the dual beam and a day in the TEM), this experiment should be repeated.

Milling sessions of the original paper were performed on a demo instrument at Zeiss (Oberkochen). We are happy to report that we performed additional sessions on a separate second instrument that has just been installed in Zürich. These sessions were successful and are represented as sessions H and I in Figure 2. Our data suggest that sessions with yeast samples and high numbers of lamellae are indeed possible.

4) If there are clear published protocols for tungsten deposition, the authors should cite them, otherwise a slightly longer explanation is warranted both for users with the same instrument and others trying to adapt the process to the tools available to them.

Additional information has been provided in the text (subsection “Sample coating”).

5) If needed, could each lamella be independently adjusted for depth? Some samples have a lot of variability and the parameters chosen can widely differ. This also goes for the height of the initial boxes. The authors always use 10 μm milling depth. Was this value an upper bound of what the depth expected their samples to have? Would a smaller value lead to lamellae not breaking through, getting curtains, etc? In other words, how should the users select this value?

Yes, the milling depth for each lamella can be independently adjusted. The 10 µm depth was used universally, as we knew that with this value we would always break through the samples. It is possible to use a smaller value to speed up the milling process, with the risk of incompletely milling the sample. We did not systematically test different values and think that there is room for optimization (trade-off between reduction of milling time and incompletely milled lamellae).

Regarding the milling pattern (boxes), all aspects of the pattern can be changed. We have included this information in the text: “It is important to note that the milling parameters (milling current, pattern size, milling depth, etc.) are adjustable on-the-fly to optimize each set of patterns for the target location and desired milling strategy.”

6) In many of the papers cited by the authors that use cryo-FIB-ET, the lamellae are much wider. While the automation will be very useful and is a much needed step in the workflow, current users can get the idea that it will increase the throughput by an order of magnitude. This is partially because the lamellae generated in this report are significantly smaller than those reported in many of the papers that the authors cite.The typical lamellae width in many of those papers are at least 10-12 μm, and a similar or longer depth. So, a typical mammalian cell (and many yeast) lamellae have an area of over 150 μm^2. In contrast, the lamellae shown here (Figure 2), have areas of about half. Similarly, the total volume that is milled away with the initial patterns is much larger in mammalian cells. Mammalian cells are at least a few thousands of cubic microns in volume, cf. the volume of a single yeast cell, 50-100 μm3. Thus, a lamella containing 3-4 yeast cells as shown in this manuscript is an order of magnitude less in volume than a mammalian cell. Thus, in order to be able to compare to other work, and to adjust expectations by the reader, the authors should specify that the numbers they cite (even in their previous manual work) of up to 16 lamellae in 10 hours applies for large single bacteria or for very small sets of yeast cells. While their reporting is accurate, it would be better (and equally impressive) to compare any gains in time with respect to their previous work.

Revised according to the reviewers’ suggestion (Discussion).